# Few-nucleon system dynamics studied via deuteron-deuteron collisions at 160 MeV

I. Ciepał[1][*], K. Bodek[2], N. Kalantar-Nayestanaki[3], G. Khatri[4], St. Kistryn[2], B. Kłos[5],
A. Kozela[1], J. Kuboś[1], P. Kulessa[6], A. Łobejko[5], A. Magiera[2], J. Messchendorp[3],
I. Mazumdar[7], W. Parol[1], R. Ramazani-Sharifabadi[3,8], D. Rozpędzik[2], I. Skwira-Chalot[9],
E. Stephan[5], A. Wilczek[5], B. Włoch[1], A. Wrońska[2], and J. Zejma[2]

**1** Institute of Nuclear Physics, PAS, PL-31342 Kraków, Poland
**2** Institute of Physics, Jagiellonian University, PL-30348 Kraków, Poland
**3** KVI-CART, University of Groningen, NL-9747 AA Groningen, The Netherlands
**4** Department of Physics and Astronomy, Northwestern University, Evanston, IL 60208, USA
**5** Institute of Physics, University of Silesia, PL-41500 Chorzów, Poland
**6** Forschungszentrum Jülich, Institut für Kernphysik, D-52428 Jülich, Germany
**7** Tata Institute of Fundamental Research, Mumbai 400 005, India
**8** Department of Physics, University of Tehran, Tehran, Iran
**9** Faculty of Physics University of Warsaw, PL-02093 Warsaw, Poland

⋆ izabela.ciepal@ifj.edu.pl

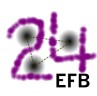 *Proceedings for the 24th edition of European Few Body Conference,
Surrey, UK, 2-6 September 2019*

## Abstract

**Four nucleon scattering at intermediate energies provides unique opportunities to study effects of the two key ingredients of the nuclear dynamics, the nucleon-nucleon P-wave (NNP-wave) and the three-nucleon force (3NF). This is possible only with systematic and precise data, in conjunction with exact theoretical calculations. Using the BINA detector at KVI Groningen, the Netherlands, a rich set of differential cross section of the $^2$H($d$, $dp$)$n$ breakup reaction at 160 MeV deuteron beam energy has been measured. Besides the three-body breakup, also cross sections of the $^2$H($d$, $^3$He)$n$ proton transfer reaction have been obtained. The data are compared to the recent calculations for the three-cluster breakup.**



## 1 Introduction

Four-nucleon (4N) system has many advantages over three-nucleon (3N) one what makes it a perfect laboratory to study nuclear forces. In particular, 4N systems reveal higher sensitivity to the effects of NN P-waves and 3NF than the ones composed of 2 and 3 nucleons. Moreover,

it is a simplest environment where 3NF in channel of total isospin T=3/2 can be investigated. 4N scattering above the breakup threshold is very difficult for theoretical treatment and currently the calculations are limited only to a few reactions. The difficulties lie in the system complexity: 4N continuum contains resonances, many input and output channels exist which are coupled and involve different isospin states. However, these are also features which make the 4N system ideal tool to study nuclear interaction and various theoretical approaches.

Recently, there has been a rapid progress in 4N accurate calculations mostly due to work of three groups: Pisa, Grenoble-Strasbourg and Lisbon-Vilnius. The Pisa group has performed accurate numerical calculations for low-energy $p-^3$He elastic scattering using hyperspherical harmonics expansion method [1, 2]. The Grenoble-Strasbourg group uses the Faddeev-Yakubovsky equations in configuration space [3, 4], but the calculations are limited so far to processes up to $n-^3$He threshold (the elastic $n-^3$He and charge-exchange $p-^3$H channels) [4, 5]. Currently, only the Lisbon-Vilnius group calculates observables for multi-channel reactions above the breakup threshold [6–8], and with the Coulomb force included. They use the momentum space equations of Alt, Grassberger and Sandhas (AGS) [9] for the transition operators. The above methods are described in the review article [10].
Comparisons between the calculations and the experimental data revealed several discrepancies, for example in $p-^3$He elastic scattering. There exist accurate measurements for unpolarized cross section [11–13], proton analyzing power $A_y$ [10, 13, 14], and other polarization observables [15]. The calculations performed with various NN interactions revealed discrepancy between theory and experiment for $A_y$ [13, 16–19]. This discrepancy is very similar to the well known "$A_y$ puzzle" in $N-d$ scattering.

Recently, the calculations have been extended for higher energies, above the four-cluster breakup threshold, up to an energy of 35 MeV. The following models were utilized in the calculations: CD Bonn [20] and Argonne V18 [21] potentials, INOY04 (the inside-nonlocal outside-Yukawa) potential by Doleschall [22], potential derived from ChPT at next-to-next-to-next-to-leading order (N3LO) [23], the two-baryon coupled-channel potential CD Bonn+$\Delta$ [24]. The last model yields effective three- and four-nucleon forces [6], but their effect is of moderate size at most. The sensitivity to the force model in the energy range studied reached 30% in the cross section minimum. The predictions have been made for observables in $p$-$^3$He [25], $p$-$^3$H, $n$-$^3$He [7] elastic scattering and transfer reactions. The first estimate calculations at higher energies have been performed in the so-called single-scattering approximation (SSA) for the $d+d \rightarrow d+p+n$ three-cluster breakup and elastic scattering [26]. The calculations are performed in two versions. The first one, the so-called one-term (*1-term*) SSA, refers to a situation in which the target deuteron breaks due to its proton interaction with the deuteron beam. The second one, the so-called four term (*4-term*) SSA, on top of the *1-term* SSA contains other three contributions, one of them corresponding to the case in which not the target proton but the neutron interacts with the beam deuteron. Two further contributions arise from exchanging the target and beam deuterons, i.e., they correspond to the breakup of the beam deuteron. Recent progress in calculations for $d+d$ system is presented in Refs. [8, 26, 27].

Experimentally, 4N systems are mostly studied in $p-^3$He and $d-d$ reactions [28–33], but in general the world database on 4N scattering at medium energies is very scarce and limited to very narrow phase space. The situation is even worse for the breakup channels [34–38]. The new-generation data covering large phase space were measured at KVI at 130 [35, 39] and 160 MeV [37, 38].

With the two deuterons in the initial state, in addition to the simple elastic scattering process, several reactions with a pure hadronic signature can occur:

1. neutron-transfer: $d+d \rightarrow p+^3$H,

2. proton-transfer: $d+d \rightarrow n+^3$He,

3. three-body breakup: $d+d \rightarrow d+p+n$,

4. four-body breakup: $d+d \rightarrow p+p+n+n$.

In this article, measurements of the differential cross section at 160 MeV beam energy for the elastic scattering and proton transfer will be presented along with three-body breakup around the quasi-free scattering (QFS) kinematical regime.

## 2   BINA detection system

The experiment was performed with the use of the Big Instrument for Nuclear Polarization Analysis (BINA) installed at Kernfysisch Versneller Instituut (KVI) in Groningen, the Netherlands. BINA is a $4\pi$ detector which was designed to detect charged particles produced in few nucleon reactions at medium energies of up to 200 MeV/nucleon. The goal of the experiment reported here was to measure differential cross section for all the channels discussed above. In the experiment, the deuteron beam of energy of 160 MeV and of very low current (about 5 pA) was impinging on a liquid deuterium target. Under such conditions the rate of accidental coincidences was minimized.

The detector is divided into two main parts, the forward Wall and the backward Ball. The forward Wall is composed of a three-plane Multi-Wire Proportional Chamber (MWPC), to reconstruct particles trajectories, and an array of $\Delta E$-$E$ telescopes to perform particle identification. The thick stopping $E$-detector, is used to measure the energy of particles. The forward Wall covers polar angles, $\theta$, in the range of 10°-35° and the full range of azimuthal angles. The backward Ball registers charged particles scattered at polar angles in the range of 40° to 165° with almost full azimuthal angle coverage. A detailed description of the BINA detection setup is presented e.g. in Ref. [33].

## 3   Results

The differential cross section for the $^2$H($d,dp$)$n$ three-body breakup at 160 MeV has been obtained for 441 proton-deuteron kinematical configurations. Polar angles $\theta_d$ and $\theta_p$ are varied between 16° and 29° with the step of 2° and proton-deuteron relative azimuthal angle $\varphi_{dp}$ is taken in the range from 20° to 180°, with the step size of 20°. The experimental results are integrated in the ranges of $\Delta\theta_p = \Delta\theta_d = 2°$ and $\Delta\varphi_{dp} = 10°$. The proton-deuteron coincidence yield has been projected onto theoretical point-like kinematics defined with $\theta_d$, $\theta_p$ and $\Delta\varphi_{dp}$ and grouped in 4 MeV $S$-bins, where $S$ is the arc-length along the kinematical curve. An example of such kinematics is presented in Fig. 1 in the left panel.   The second introduced variable, $D$ denotes the distance of each experimental point $(E_d, E_p)$ from the point-like kinematics see Fig. 1. The cross section has been calculated in a function of $S$. The gaussian was fitted to the $D$-distributions and the events have been integrated in a $D_a$-$D_b$ range (see Fig. 1, righ panel) corresponding to distances of $-3\sigma$ and $+3\sigma$ from the maximum of the fitted peak. The events have been corrected for the acceptance and efficiency and normalized to the $d-d$ elastic scattering cross section measured at two energies of 130 and 180 MeV [32]. For more details see Refs. [33, 37, 38]. The data have been compared to the recent theoretical predictions which are available only for configurations around the QFS region, in the so-called single-scattering approximation (SSA) for the three-cluster breakup [26], see also Sec. 1. The calculations have been performed using the CD Bonn, AV18 and CD Bonn + $\Delta$ potentials in *1-* and *4-term* versions of the SSA calculations. In the investigated part of the phase space, corresponding to the forward part of the BINA setup, the breakup channel is dominated by

the quasi-free scattering of the deuteron beam on proton from the deuteron target. This effect can be seen in Fig. 3, left panel, which presents the experimental cross sections integrated over the $S$ arc-length with the analogously treated theoretical predictions. The cross section is strongly peaked in the region of quasi-free scattering and almost two orders of magnitude smaller for other geometries. The data have been compared to the available SSA calculations at $\varphi_{dp}=140°$, $160°$ and $180°$. One can also notice in Fig. 3 that with $\varphi_{dp}$ approaching to $180°$ the data description is getting better. Among all coplanar geometries, the ones with larger polar angles ($\theta_p \geq 24°$ and $\theta_d \geq 24°$) are better described by theory, as expected from SSA.

In the previous analyses of the $ppn$ channel [40–43] the standardized $\chi^2$ was used to compare the data with various exact calculations. Owing to the fact that currently only approximate calculations are available and discrepancies between the data and theories are quite large, a so-called $A$-factor instead of $\chi^2$ was used. The $A$-factor was introduced in Ref. [38,44] and is defined as follows:

$$A \equiv \frac{1}{N} \sum_{i=1}^{N} \frac{|\sigma_i^{exp} - \sigma_i^{th}|}{\sigma_i^{exp} + \sigma_i^{th}},\tag{1}$$

where the sum runs over the number of data points in a given bin of a given variable. The $A$-factor has the advantage of its quite simple interpretation. Values of the $A$-factor belong to the interval [0, 1], where zero means a perfect agreement between the data and calculations and with the deterioration of the agreement the $A$-factor approaches one. To investigate the quality of the data description by the model, the data have been sorted with respect to the neutron-spectator energy $E_n$. In the investigated part of the phase space, corresponding to the so-called "around" $dp$-QFS region [38], the neutron energy (or momentum) is suitable variable to select the QFS process (besides the kinematical relations). In general one can conclude that at the lowest $E_n$ the *1- term* calculations perform better than those with *4-term*, see Fig. 3, right panel. For $E_n < 10$ MeV the *1- term* calculations describe the data well. For higher $E_n$ the agreement between the experimental and calculated cross sections deteriorates, but the *4- term* calculations stay closer to the data than the *1- term* ones. At the highest available $E_n$ the $A$-factor evaluated for all calculations has values close or equal to one which means a

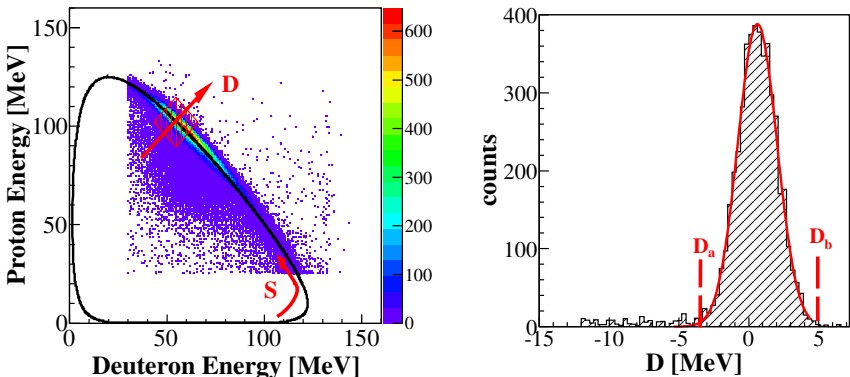

Figure 1: (Color online) *Left panel:* $E_p$ *vs.* $E_d$ coincidence spectrum of the proton-deuteron pairs registered at $\theta_d = 24° \pm 1°$, $\theta_p = 28° \pm 1°$, $\varphi_{dp} = 180° \pm 5°$. The solid line shows a 3-body kinematical curve calculated for the central values of the angular ranges. Variables: arc-length $S$ and distance from kinematics $D$ are presented in a schematic way. *Right panel:* The projection of events belonging to one $\Delta S$ bin onto the $D$-axis. A Gaussian distribution was fitted in the range of $D$ between $D_a$ and $D_b$, corresponding to distances of $-3\sigma$ and $+3\sigma$ from the fitted peak position.

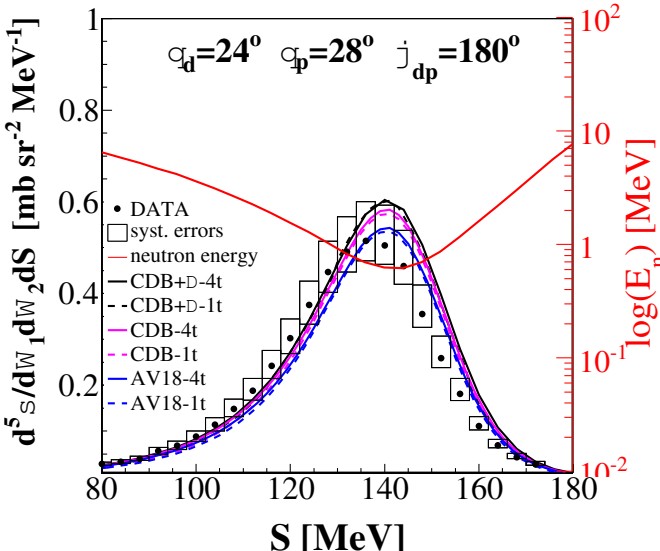

Figure 2: (Color online) Differential cross section as a function of the kinematical variable *S* for the angular configuration specified in the panel. The experimental points are marked with black dots with total systematic errors depicted as empty boxes. Various lines represent the theoretical predictions calculated at the central values of the defined angular bins. The black lines refer to the SSA calculations based on the CD Bonn + $\Delta$ (CDB + $\Delta$) potential: solid with *4-term* (4t) and dashed with *1-term* (1t). Solid and dashed magenta (light gray) and blue (dark gray) lines represent the similar set of the calculations but for the CD Bonn and AV18 potentials, respectively. The solid red line (upper line) presents the dependence of the spectator neutron energy ($E_n$) on the *S* variable. For details see also Ref. [38].

failure of the theoretical description, as expected from the model assumptions [26].

Additionally to the breakup cross sections, differential cross section for the proton-transfer $d+d \rightarrow n+^3$He has been obtained [33]. The BINA cross section is presented together with previously measured data in a function of square of four-momentum transfer in Fig. 4. The $q^2$ distribution follows the trend observed at higher energies obtained in proton and neutron transfer reactions. Currently, there is no theoretical predictions available to compare with the data.

# 4 Further studies at higher energies

The studies can be further extended to energy of 350 MeV on the basis of the *dd* scattering data collected with the WASA@COSY detector [45].

The WASA (Wide Angle Shower Apparatus) detection system operated at the COSY (COoler Synchrotron) accelerator in FZ-Jülich and consists of four main components: Central Detector (CD), Forward Detector (FD), Pellet Target Device and the Scattering Chamber. The CD surrounds the interaction region and is used for detection of neutral and charged particles. It contains four detectors: Mini Drift Chamber, Plastic Scintillator Barrel, Scintillator Electromagnetic Calorimeter and Super Conduction Solenoid. The FD covers the region of the polar angles from 3° to 18°. It consists of a set of plastic scintillators for the charged hadron identification, track reconstruction and energy measurement. It contains different types of the detectors: Forward Window Counter, Forward Proportional Chamber and Forward Trigger

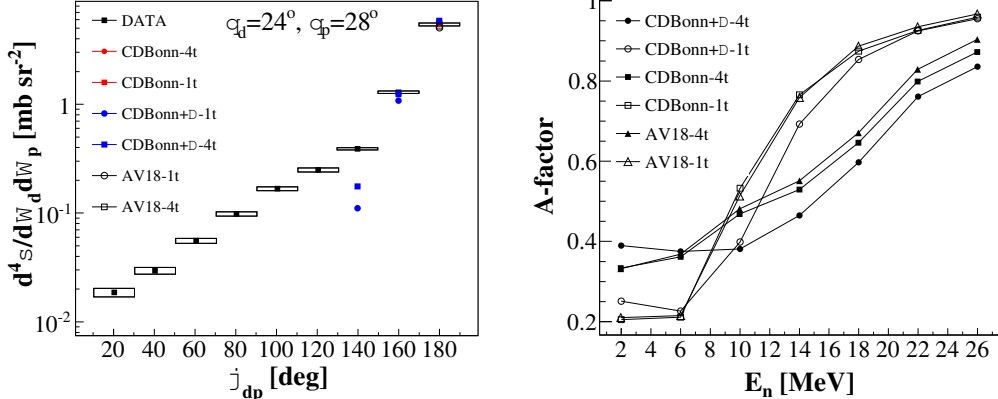

Figure 3: (Color online) *Left panel:* Diffferential, but integrated over $S$, cross section presented as a function of the relative azimuthal angle $\varphi_{dp}$, for a given $\theta_d$, $\theta_p$ combination as indicated in the panel. The data points are compared with the results of calculations based on pure CD Bonn, Argonne V18 and CD Bonn+$\Delta$ potentials for *1-term* (*1t*) and *4-term* (*4t*), as described in the legend. The boxes around the experimental points represent the total systematic errors. *Right panel:* Quality of the description of the cross-section data with various theoretical predictions (defined in the legend), expressed as a dependence of the $A$-factor on the neutron energy $E_n$. The bin size was chosen to be $\pm 2$ MeV. Lines connecting points are used to guide the eye. For details see also Refs. [37,38].

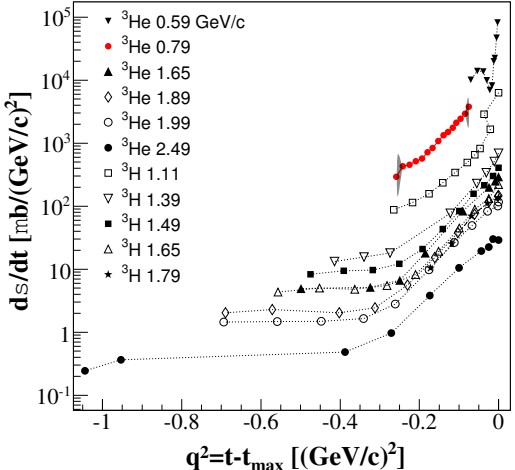

Figure 4: (Color online) Differential cross sections for transfer reactions in *d-d* scattering $q^2$ [29,30]. The data from the BINA experiment at 0.79 GeV/c are shown as red points. The dark band represents systematic errors. For more details see Ref. [33].

Hodoscope.

It would be very interesting to check if at almost two times higher energy the SSA calculations will perform better than at 160 MeV. Sample calculations for FD-FD deuteron-proton geometries are presented in Fig. 5. The red lines represent the neutron energy and at its minimum the QFS process can be investigated.

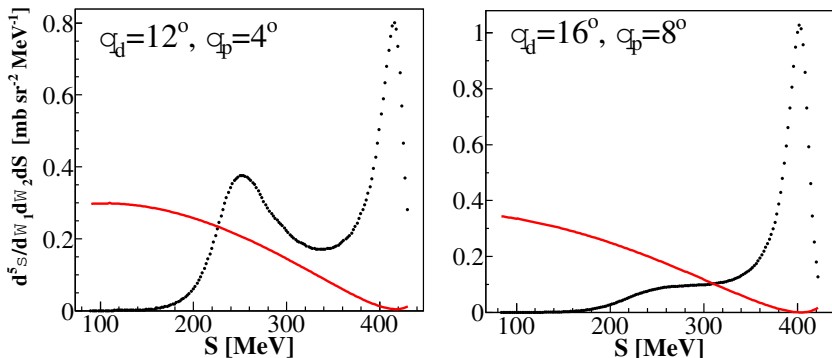

Figure 5: (Color online) Differential cross section for the angular configuration specified in the figures. The black points refer to the SSA calculations based on the CD Bonn + $\Delta$ potential in the *4-term* version. The solid red line present the dependence of the spectator neutron energy ($E_n$) on the $S$ variable.

## 5 Conclusions

The differential cross-section distributions for the three-body $^2$H($d$, $dp$)$n$ breakup reaction have been obtained for 441 proton-deuteron geometries at 160 MeV deuteron beam energy. The analyzed part of the phase space covers a wide kinematical region around the $dp$-QFS. The cross sections have been compared to the calculations based on the single-scattering approximation for 4N systems at higher energies [26]. The system dynamics is modeled with AV18, CD Bonn and CD Bonn+$\Delta$ potentials. The calculations are still not exact, but they provide a correct order of magnitude for the cross section close to the QFS region.

Additionally to the breakup cross sections, the differential cross section for the $dd \rightarrow n^3$He proton transfer reaction have been obtained. In this case, however, no calculations exist, but the data can be used to validate the future theoretical findings. They supplement the existing database in the poorly known region of intermediate energies (beam momentum below 1 GeV/c).

There is possibility to study the three-body breakup at 350 MeV with the $dd$ scattering data collected with the WASA@COSY detector to test the SSA calculations at higher energy.

## Acknowledgements

This work was supported by the Polish National Science Center under Grants No. 2012/05/E/ST2/02313 (2013-2016) and No. 2016/21/D/ST2/01173 (2017-2020), and by the European Commission within the Seventh Framework Program through IA-ENSAR (Contract No. RII3-CT-2010-262010).

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
