# Peer review of "Few-Nucleon System Dynamics Studied via Deuteron-Deuteron Collisions at 160 MeV"

_SciPost Physics Proceedings, doi:SciPost Phys. Proc. 3, 018 (2020)_

## Round 1 · Referee Report · Paul Stevenson (Referee 1) · 2019-12-11

Report

Report on Few-Nucleon System Dynamics Studiedvia Deuteron-Deuteron Collisions at 160 MeV by Ciepal et al. This paper shows results of 2H(d,dp)n experiments at 160MeV deuteron beam energy as carried out at KVI Groningen. The experiments aim to study particular effects which are best studied in 4-body systems (e.g. NN P-waves, aspects of 3NF) and compare to recent theory which has been able to make sophisticated calcualtions of such 4-body systems recently. The introduction includes a concise summary of the theoretical status of relevant calculations, highlighting the need for new experimental data. Section 2 gives a brief but sufficient description of the experimental setup, with a reference to a full desription The results section shows some sample results, including a comparison of the differential cross section to a range of theoretical calculations for a particular kinematic setup, showing pretty good agreement. Further details (e.g. as a function of angle) show up where the remaining discrepancies in the theoretical approaches lie in comparison with data. The paper then gives an interesting look forward to possible future experiments. Overall it is well-written and an interesting document of the work presented at the conference. It can be published as is, though one typo was noted as shown in the requested changes section of the report. This can be corrected at the proof stage

Requested changes

caption of figure 3 includes three 'f's in 'Diffferential'

---

## Editorial Decision

published